# The Effects of Ethanol and Acetic acid on Behaviour of Extranidal Workers of the Narrow-Headed Ant *Formica exsecta* (Hymenoptera, Formicidae) during a Field Experiment

**DOI:** 10.3390/ani13172734

**Published:** 2023-08-28

**Authors:** Julita Korczyńska, Anna Szczuka, Julia Urzykowska, Michał Kochanowski, Neptun Gabriela Andrzejczyk, Kacper Jerzy Piwowarek, Ewa Joanna Godzińska

**Affiliations:** 1Laboratory of Ethology, Nencki Institute of Experimental Biology PAS, Ludwika Pasteura 3, PL 02-093 Warsaw, Poland; j.korczynska@nencki.edu.pl (J.K.); a.szczuka@nencki.edu.pl (A.S.); j.urzykowska@student.uw.edu.pl (J.U.); 2Faculty of Biology, University of Warsaw, Ilji Miecznikowa 1, PL 02-096 Warsaw, Poland; 3Botanic Garden, University of Warsaw, Aleje Ujazdowskie 4, PL 00-478 Warsaw, Poland; mj.kochanowski@student.uw.edu.pl; 4Department of Animal Physiology, Institute of Functional Biology and Ecology, Faculty of Biology, University of Warsaw, Ilji Miecznikowa 1, PL 02-096 Warsaw, Poland; g.andrzejczy@student.uw.edu.pl (N.G.A.); k.piwowarek@student.uw.edu.pl (K.J.P.)

**Keywords:** ethanol, acetic acid, behaviour, locomotion, self-grooming, aggressive behaviour, ants, Formicidae, *Formica exsecta*

## Abstract

**Simple Summary:**

Alcoholism (addiction to ethanol, a common alcohol present in our beverages) is among the most important problems encountered in the domain of human mental health. The research devoted to behavioural effects of exposure to/consumption of ethanol is carried out largely with the help of animal models that include not only vertebrates such as rats and mice, but also insects, in particular fruit flies and honeybees. The effects of ethanol on ant behaviour remain, however, little known. In the present field study, we investigated the behaviour displayed by workers of the narrow-headed ant (*Formica exsecta*) in the vicinity of cotton pads soaked either in water or in water solutions of ethanol or acetic acid. Both ethanol and acetic acid induced significant modifications of ant behaviour, influencing various aspects of ant locomotion, exploratory behaviour, self-grooming behaviour, and aggressive social behaviour. We confirmed that acetic acid is aversive for the ants, but ethanol induces their interest. We also found out that field studies may discover more effects of experimental compounds than laboratory ones, as the tested animals may escape from aversive substances. Our present results also broadened the general knowledge about behavioural responses to ethanol and acetic acid encountered in animals.

**Abstract:**

Ethanol addiction belongs to the most important problems encountered in the domain of human mental health. The research on the behavioural effects of exposure to/consumption of ethanol are investigated largely with the help of animal models that also include insects, mainly fruit flies and honeybees. The effects of ethanol on ant behaviour remain, however, little known. In the present field study, we investigated the behaviour of workers of the narrow-headed ant (*Formica exsecta*) displayed in the vicinity of cotton pads soaked in water or in water solutions of ethanol or acetic acid during 5 min tests (*n* = 30 tests in each group). Both ethanol and acetic acid induced significant modifications of ant locomotion, exploratory behaviour, self-grooming behaviour, and aggressive social behaviour. We confirmed that acetic acid is aversive for the ants, but ethanol enhances their exploratory behaviour. We also found out that field studies may document more types of responses to experimental compounds than laboratory ones, as the tested animals may also escape from aversive substances. Our findings documented a wide spectrum of behavioural effects of exposure to ethanol and acetic acid in a highly social animal species and broadened the general knowledge about behavioural responses to these compounds encountered in animals.

## 1. Introduction

### 1.1. The Role of Animal Models in the Research on Human Mental Health and Mental Disorders

Experimental research carried out to shed more light on factors involved in the mediation of various aspects of human mental health and on biological correlates of mental disorders is carried out largely with the help of animal models [1,2,3]. Model animals used in that research include not only species relatively closely related to humans, such as non-human primates [4], but also laboratory rodents (mainly mice and rats, also genetically modified ones) [5,6], other vertebrates much less closely related to humans such as zebrafish (*Danio rerio*) [7,8], and, last but not least, various invertebrates, in particular solitary and social insects [6,9,10,11,12,13,14].

### 1.2. Insect Models Used in the Research on Drug and Ethanol Addiction

Drug addiction (including ethanol addiction) belongs to the most important problems encountered in the domain of human mental health [2,15]. Insect species most frequently used in the research caried out to broaden our knowledge about the biological roots of drug addiction (including the experiments carried out to shed more light on behavioural and physiological effects of consumption of/exposure to ethanol) include, in particular, two species: the fruit fly (*Drosophila melanogaster*) [6,10,11,13,14,16,17] and the honeybee (*Apis mellifera*) [9,12,13].

### 1.3. Methods of Administration of Ethanol to Insects

The research carried out to shed more light on the role of ethanol in the mediation of behaviour of members of both these model insect species has already documented a wide range of effects. In a large portion of these studies, the tested insects received ethanol by means of oral treatment; it was added to a sucrose solution, and sometimes also to other food. Such a method of administration of ethanol was applied in the experiments carried out with the use of both *Drosophila* fruit flies [18,19,20] and honeybees [21,22,23,24,25,26,27,28,29,30,31,32,33,34]. However, in other studies the tested insects were exposed to vapours of ethanol and sometimes also to vapours of other control compounds such as water or ethyl acetate. This method of administration of ethanol was also applied in the case of both *Drosophila* fruit flies [20,35,36,37,38,39,40,41,42] and honeybees [43,44,45,46,47]. Yet another method of exposure to ethanol used in a study with *Drosophila* larvae consisted of 20 min treatments during which the tested insects were kept in a Petri dish containing either water or ethanol solution [48].

### 1.4. Interrelationships between the Consumption of/Exposure to Ethanol and Behaviour, Cognitive Proceses and Depression-like States Encountered in Insects

Behavioural effects of ethanol documented by all that research included, above all, a wide spectrum of modifications of locomotion (both walking and flight) and, in particular, sedation, hyperactivity, and modifications of turning behaviour and postural control (*Drosophila* flies: [34,35,36,38,39,40,41,42]; honeybees: [21,28,33,43,44,45,46,47]). Many studies were also focused on the effects of ethanol on cognitive processes encountered in the tested insects including various types of learning and complex decision processes (*Drosophila* flies, adults and larvae: [18,48] and honeybees: [21,22,25,26,29,31]).

Interesting results were also brought about by the research carried out to better understand the interrelationships between, on the one hand, stress and depression-like states, and, on the other hand, consumption of/exposure to ethanol. Thus, sexual deprivation was found to increase ethanol intake in *Drosophila* males [49]. Similarly, sleep deprivation and social isolation were shown to increase resistance to ethanol sedation in sexually mature *Drosophila* females. Recovery of sexually mature *Drosophila* females from ethanol-induced sedation was also less affected by stress than recovery of immature females and both immature and mature males [42].

Consumption of/exposure to ethanol was also found to enhance various forms of insect aggressive behaviour. Such effects were documented in both *Drosophila* fruit flies [20] and in honeybee workers tested by means of various behavioural bioassays [24,30]. However, in another study [21], ethanol consumption did not influence stinging behaviour of the tested honeybee workers, and in yet other one [31] it was followed by the increase of sting extension response (SER) thresholds, and, consequently, decreased frequency of defensive behaviour related most probably to analgesic effects of ethanol consumption. Lastly, yet another study [43] was carried out to investigate an inverse effect: the influence of aggressiveness of honeybee workers on their sensitivity to ethanol. In that experiment, upon exposure to vapours of ethanol, workers from a high-defensive colony displayed characteristic signs of ethanol-induced sedation significantly faster than workers from a low-defensive (“gentle”) colony.

Some studies also documented the effects of ethanol on friendly social contacts between honeybee workers. In particular, consumption of ethanol solutions offered to the honeybee foragers at the feeding site was found to influence several patterns of social behaviour displayed by them upon their return to the hive, including a reduction of their waggle dance activity and increased occurrence of tremble dance, food exchange, and self-cleaning behaviour [27]. Relatively low doses of ethanol administered in food were also found to influence many aspects of behaviour of honeybee workers tested in laboratory bioassays during which a dyad of nestmates from the same colony was confined together in a small container. Although inebriated bees were more likely to display open-mandible threats, they were as willing as the control (sucrose-fed) bees to engage in trophallaxis (exchange of liquid food and various active and signal compounds during a mutual contact of mouthparts; see [50]). Interestingly, a trophallactic contact with a donor previously fed with ethanol was followed by the state of inebriation also in the receiver of the liquid food, which evidently still contained ethanol [30].

### 1.5. Preference for Ethanol as an Evolutionary Adaptation

We also should remember that preference for ethanol shown by insects may act as an important evolutionary adaptation. Thus, females of *Drosophila* fruit flies were found to respond to contacts with endoparasitoid wasps by switching to laying eggs in ethanol-laden food sources. Food containing ethanol protects hatched *Drosophila* larvae from infection. Interestingly, that response evolved multiple times in the fruit flies of the genus *Drosophila* [51]. Further research devoted to that phenomenon [52] revealed that exposure of the *Drosophila* fruit flies to predatory wasps leads to inheritance of a predisposition for ethanol-rich food for five generations. Exposure of *Drosophila* larvae to moderate, naturally occurring ethanol concentration of 4% was also found to exert a beneficial effect on larval survival [53].

### 1.6. Ants as Subjects in the Research Carried out in the Field of Behavioural Neuroscience and Neuropsychopharmacology

Surprisingly, ants, social insects closely related to the honeybees, were so far much less frequently used in the research carried out in the field of behavioural neuroscience and neuropsychopharmacology. However, such studies also yielded interesting results. In particular, two closely related species of the red wood ants from the genus *Formica*, *Formica polyctena* and *Formica rufa*, were already used in a relatively large number of both classical and recent studies carried out to investigate neurochemical processes involved in the mediation of aggressive behaviour. These studies yielded a wide spectrum of important results and, in particular, discovered and documented many striking similarities between neurochemical processes mediating aggressive behaviour in ants and in vertebrates including humans (for the reviews, see [54,55]). Neurochemical processes involved in the mediation of various subcategories of friendly social behaviour were also studied in various ant species, in particular in the ants of the genus *Formica* and in the carpenter ants of the genus *Camponotus* (for the reviews, see also [54,55]). Nevertheless, we should bear in mind that human and ant social behaviour should be compared in a very careful way to draw valid parallels [56].

### 1.7. Field Versus Laboratory Studies of the Effects of Ethanol on Insect Behaviour and the Main Aims of the Present Research

So far, experimental research carried out to investigate the responses of various insects to ethanol and the effects of consumption of that compound or exposure to its vapours was carried out largely in laboratory conditions, and only a very small number of these studies was also made partly or entirely in the field [24,27,33]. Therefore, the main aim of the present study was to investigate the effects of ethanol on ant behaviour in field conditions by comparing them with the effects of exposure to water and to acetic acid presented to the ants in exactly the same manner, on small cotton pads soaked in the tested solution. Water was chosen as one of the controls as it was relatively neutral from the point of view of olfactory perception, and acetic acid was chosen as the second control because, similarly to ethanol, it provided relatively strong olfactory stimulation. Such an experimental design was chosen to allow us to document a wide spectrum of both specific and non-specific behavioural effects induced by these compounds. We were also aware of the fact that closely similar effects of exposure to both ethanol and acetic acid will rather have to be attributed to non-specific modifications of behaviour induced by exposure to strongly smelling olfactory stimuli.

An important advantage of the experimental methods applied in our present study was also related to the fact that a field experiment allows the observers to record behaviour displayed by the tested ants in conditions in which they can escape from the vicinity of experimental objects and/or can avoid more prolonged contacts with the experimental compound(s). In laboratory experiments, escape/avoidance of the tested insects from exposure to vapours of test compounds is impossible, as such exposure takes place in closed containers [20,35,36,37,38,39,40,41,42,43,44,45,46,47]. Therefore, we expected that our field study will yield findings more strongly related to behavioural processes taking place in the natural environment of the tested animals than findings obtained in laboratory experiments.

### 1.8. Ant Species

We used as subjects ants from the species *Formica exsecta* (Hymenoptera: Formicidae, subfamily Formicinae), also known as the narrow-headed ant or excised ant. These mound-building ants belong to the same genus *Formica* as the common red wood ants from the group *Formica rufa*, but their mounds are smaller and they are known mostly for their sophisticated social behaviour. In particular, they often form huge polydomous (polycalic) colonies composed of large complex systems of numerous interconnected nests [57,58,59,60,61,62,63,64,65]. One of such colonies found in Romania was claimed to represent the largest polydomous system of *Formica* ants discovered in Europe [65]. The narrow-headed ants belong thus to the best known highly social ant species [57,58,59,63,65].

## 2. Materials and Methods

### 2.1. Field Site

The experiment was carried out in the field, within a large polydomous colony of *Formica exsecta* situated in Falenica, a part of Wawer, one of the districts of Warsaw (Poland) located on the right bank of the Vistula River, in the far southeastern corner of the city (GPS coordinates of the central part of that colony: 52.154789 N, 21.181835 E). The studied colony consisted of a system of numerous nests taking the form of relatively small mounds (usually about 10–15 cm high) composed of tiny pieces of plant material, and covered the surface of about 400–500 m^2^ within an open grassland area overgrown with tall (30–50 cm) grass and partly with young trees (about 4–5 m high), mostly pine with a small addition of birch and oak. The nests used in our experiment (in total 8 mounds) were all situated in dry, sunny places at the southern side of clumps of young pine trees.

### 2.2. Experimental Compounds and Their Concentrations

During each test, we observed the behaviour of extranidal workers of *Formica exsecta* displayed in the vicinity of a small (1 cm in diameter) cotton pad soaked in 1 mL of water (distilled water for injection), aqueous solution of ethanol (96%), or aqueous solution of acetic acid (10%). Concentrations and volume of these solutions were chosen on the basis of pilot laboratory tests during which single extranidal workers (foragers) of *Formica exsecta* were exposed to vapours of water, ethanol, or acetic acid in inhalation chambers similar to the ones used to study the effects of exposure to ethanol in fruit flies [20,35,36,37,38,39,40,41,42] and in honeybees [43,44,45,46,47]. Some of these pilot tests brought about highly undesirable effects. Thus, 20 min exposure to vapours of ethanol produced by 1.6 mL of 96% ethanol solution and 15 min exposure to vapours produced by 1 mL of such solution was followed by severe worker mortality: 7 out of 10 workers subjected to these treatments were found dead on the next day. However, 5 min exposure to vapours emitted by 1 mL of water, 96% aqueous solution of ethanol, or 10% aqueous solution of acetic acid did not entail such undesirable effects. All the ants exposed to these treatments (*n* = 10 in each group, in total 30 ants) were alive and in good physical form on the next day.

Therefore, exactly the same treatments were used by the members of our team in a laboratory experiment in which single foragers of the narrow-headed ant (*Formica exsecta*) were first placed in an inhalation chamber and subjected to 5 min exposure to vapours of water, 96% ethanol, or 10% acetic acid, and immediately afterwards were transferred to an experimental arena containing a nestmate worker trapped in an artificial snare and several small inanimate objects, and observed there during a 15 min long behavioural test during which their behaviour was videorecorded. Each experimental group consisted of 32 workers, but we analysed only the recordings of behaviour of 90 (3 × 30) workers, as no ant died either in the inhalation chamber, or during the subsequent behavioural test, and no test had to be discarded from the analysis because of important motor disturbances of the tested ant. The analysis of behaviour of the ants tested in that experiment is still not complete, but we can already state that both experimental compounds, and in particular ethanol, induced modifications of a wide spectrum of behaviour patterns, including various aspects of locomotion, exploratory behaviour, self-grooming behaviour, and social behaviour. Aggressive behaviour (open-mandible threats and biting behaviour directed to various elements of physical environment) was relatively frequently observed only in the ants treated with ethanol. From the point of view of our present study, the results of that first laboratory experiment strongly suggested that the concentrations of ethanol and acetic acid used safely in almost one hundred behavioural tests can be equally safely used in other behavioural experiments with ants.

### 2.3. Tests

Experimental objects (1 cm in diameter cotton pads soaked in 1 mL of either water or aqueous solution of ethanol or acetic acid with the use of a disposable syringe) were presented to the ants on naturally occurring small (at least 5 cm in diameter) bare patches of ground situated relatively close to ant mounds, at a distance of 15–80 cm from the outer border of the closest mound. The experimental solutions were kept by us in small, closed bottles, and we have no reasons to suspect that their concentrations did not retain stability during the whole session of field tests.

Prior to each test, we first selected ant nests with relatively high levels of worker activity taking place in their vicinity, and then we checked the intensity of ant traffic at the bare patch of ground chosen for the proper test. To that purpose, we counted the number of ants crossing an approximately 5 cm in diameter circle in the middle of the observed patch during 1 min. The borders of that circle were left unmarked to avoid unnecessary disturbance of ant behaviour. The person that observed the activity of the ants also focused attention on various small objects (pebbles, grass blades, etc.) acting as field marks that helped him to identify the limits of that circle. The proper test was subsequently carried out at that spot only if at least 8 ants crossed that area during the preliminary 1 min observation. We chose that criterion for the choice of a particular spot for the field test, as this allowed us to expect that during the subsequent 5 min test we will be able to observe the behaviour of several tens of ants, and this in turn will provide us with a sufficient amount of behavioural data to make possible reliable and valid conclusions. If the number of the ants observed to cross that area was lower, we repeated the preliminary observation, choosing another patch of bare soil situated close to the same nest. Immediately before the start of each proper test, a freshly prepared experimental cotton pad was placed in the centre of the circular area used as the point of reference during the preliminary observation. Each test was continued for 5 min following the first contact of the ants with the test object, or, more precisely, the first instance of any behaviour included in the list of our behavioural categories. Physical contact of the ant with the experimental object was not necessary. During the subsequent 5 min period, we noted occurrence of various behavioural events classified into 16 behavioural categories, quantifying the responses of the ants to the experimental cotton pad and its surroundings and including their responses to other nestmate ants. We also recorded the latencies from the start of the test to the first observation of each behaviour pattern. Each test was carried out by two persons. The first one acted as an observer and reported the observed behaviour patterns, and the second one noted this information together with the timing of the first occurrence of each behaviour measured by means of a stopwatch activated at the start of each test. These two observers were able to manage to document all behavioural events, also in situations when more than one ant was present close to the test object or when the same ant performed a sequence of several behavioural responses.

Each series of three tests, each with a different type of experimental solution, was carried out close to the same ant mound. The tests with different compounds were always performed in the same order (water–ethanol–acetic acid), to make sure that during the initial tests with water the ants would not be exposed to any strongly smelling experimental compound, and during the tests with ethanol they would not be exposed to the smell of acetic acid. We took care to make sure that the distance between the spots at which the ants were tested with water and with one of the strongly smelling compounds (ethanol or acetic acid) was no shorter than 15 cm, and that the distance between the spots at which the ants were tested with ethanol and with acetic acid was no shorter than 50 cm. After the test each experimental cotton pad was placed in a closed plastic bag and treated as trash.

The distance between the nests used for two successive series of three tests ranged from about 1.5 m to about a dozen of metres. The same nest was never used twice during two successive series of tests. However, during the whole experiment, the surroundings of the majority of these nests were used for more than one series of the tests (up to 9 repetitions per nest). Only nests 4 and 8 were used a single time. However, no patch of bare soil was used more than once during the whole experiment. These bare patches of ground had characteristic features, and their location was easy to be remembered.

Altogether, the responses of the ants to each experimental compound were observed during 30 tests. The whole experiment consisted of 90 tests carried out on 6 days: 30th and 31st August 2022, and 6th, 7th, 8th, and 26th September 2022. The tests were carried out during the hours 10.00–17.00. We chose the days with similar weather conditions (sunny weather with at the most partial cloud cover). As workers of *Formica exsecta* tended to avoid being active in areas directly exposed to the sun, our tests were made during the periods when the sun was hidden behind clouds, in areas placed in natural demi-shadow. In some cases, we also provided artificial shadow with the help of an umbrella.

Air temperature (measured at ground level immediately after the end of each test close to the spot at which that test was carried out) showed relatively important differences, ranging from 13 °C to 28 °C. However, our experiment consisted of series of three tests, each with a different experimental compound, performed in succession close to the same nest. Therefore, the temperature measured immediately after the end of each of these tests was closely similar. This conclusion was confirmed by Friedman ANOVA analysis in which the results obtained for the tests from the same three test series were treated as dependent data. That analysis did not discover significant inter-group differences in air temperature measured immediately after the test.

### 2.4. Behavioural Categories

During our tests, we quantified the behaviour of the observed ants, taking into account the following 16 behavioural categories:(1)Ignoring the experimental object by an ant passing close to it;(2)Stopping close to the experimental object accompanied by movements of the antennae;(3)Withdrawal from the experimental object immediately after stopping close to it;(4)Bypassing the experimental object (giving it a wide berth, but within a distance no longer than 2–3 ant body lengths);(5)Jumping away from the experimental object after an antennal contact with it;(6)Antennal contact/contacts with the experimental object;(7)Nibbling of the experimental object;(8)Nibbling of the experimental object accompanied by gaster flexing, a form of threatening behaviour;(9)Nibbling at the substrate close to the experimental object;(10)Self-grooming of the antennae close to the experimental object;(11)Self-grooming of the gaster close to the experimental object;(12)Charge at a nestmate observed close to the experimental object;(13)Open-mandible threat directed at a nestmate close to the experimental object;(14)Dragging of a nestmate close to the experimental object;(15)Antennal contact/contacts with a nestmate close to the experimental object;(16)Allogrooming of a nestmate close to the experimental object.

The expression “close to the experimental object” means here “at the distance no longer than 3 body lengths of the tested ants”.

### 2.5. Statistical Analysis of the Data

Each behavioural category was quantified by three variables: the rate of occurrence of the tests during which the behaviour in question was observed, the latency from the start of the test to the first observation of that behaviour, and the number of observations of that behaviour recorded during the test.

The values of each of these variables obtained for three experimental groups were first compared by means of the overall analysis of all obtained data (2 × 3 Fisher Exact Probability Test in the case of the rate of occurrence of the tests during which the behaviour in question was observed, and Kruskal–Wallis ANOVA in the case of the other two variables). If such an analysis yielded a significant result (*p* ≤ 0.05), it was followed by three pairwise post hoc tests comparing the data obtained for specific pairs of experimental groups (2 × 2 Fisher Exact Probability Tests in the case of the first variable, and Siegel and Castellan post hoc tests for pairwise comparisons of independent data taking into account the inherent error rates accompanying multiple comparisons [66] in the case of the other two variables). We also noted all cases in which overall statistical analyses of our data and/or post hoc tests revealed a non-significant trend (0.05 < *p* ≤ 0.1).

Similarly as in the case of some of our earlier studies [67,68,69], the values of the latency from the start of the test to the first observation of the analysed behaviour were expressed as per cent of the total test time measured from the start of the test until the onset of the first bout of that behaviour, as such a variable could be calculated also for the tests during which that behaviour was absent, taking the value of 100%.

In the case of three behavioural categories (nibbling of the experimental object, self-grooming of the antennae, and charge at a nestmate), we also carried out additional analyses of the latency from the start of the test to the first observation of that behaviour, and of the number of observations of that behaviour recorded during the test in which we took into account only the tests during which the behaviour in question was expressed. In the case of the first two behavioural categories, these additional analyses were carried out by means of the Kruskal–Wallis ANOVA followed by Siegel and Castellan post hoc tests for pairwise comparisons of independent data if Kruskal–Wallis ANOVA yielded a significant result. In the case of the charge at a nestmate, we had to compare only two groups, ants tested with ethanol and ants tested with acetic acid, as that behaviour was never observed during the tests with water. Therefore, we compared these data by means of the two-tailed Mann–Whitney U test.

Only these three behavioural categories were selected for these additional analyses, as in the case of the remaining behavioural categories the behaviour was observed either on all tests (in the case of two behavioural categories, ignoring of the experimental object and antennal contact/contacts with it) or was observed so very infrequently that sample sizes taking into account only the tests during which a given behaviour was observed would be too small to expect any interesting outcomes of such an analysis.

## 3. Results

### 3.1. General Remarks

The analysis of the results of this study discovered a relatively large number of significant differences between behaviour patterns shown by extranidal workers of the narrow-headed ant (*Formica exsecta*) close to the experimental cotton pads soaked in water, aqueous solution of ethanol, and aqueous solution of acetic acid. These differences consisted mostly (but not solely) of modifications of locomotion occurring close to the experimental object (withdrawal from the experimental object and bypassing of that object). However, we also observed significant inter-group differences in the case of antennal contacts with the experimental object, self-grooming of the antennae close to that object, and aggressive responses of the ants to other nestmates present close to that object (charges at a nestmate).

The results of our study are presented as successive subchapters containing, one after another, the results obtained for all investigated behavioural categories. These categories (in total 16 ones) are presented in a logical order, starting from the behaviour patterns representing various forms of locomotion and/or modifications of locomotion, up to the behaviour patterns belonging to the more general categories of aggressive behaviour and friendly social behaviour. Each behavioural category is named in the subheading of the respective subchapter. Each subchapter contains the results of the analysis of three variables chosen to quantify each behavioural category always presented in the same order: the rate of occurrence of the tests during which the behaviour in question was observed, the latency from the start of the test to the first occurrence of that behaviour, and the number of observations of that behaviour recorded during the test. The most important results are also shown in Figures.

### 3.2. Ignoring the Experimental Object

Ignoring the experimental object by an ant passing close to it was observed during all the tests. In other words, the rate of occurrence of the tests during which that behaviour was observed was equal to one in all experimental groups, and, consequently, did not show inter-group differences.

Kruskal–Wallis ANOVA analysis of the latencies to the first observation of that behaviour recorded during the tests with different experimental objects did not discover statistically significant differences between the compared groups, or even a non-significant trend.

However, in the case of the numbers of observations of that behaviour recorded during the tests with different experimental objects Kruskal–Wallis ANOVA discovered a non-significant trend (Figure 1). The values of that variable tended to be the lowest in the case of the tests with ethanol. However, as Kruskal–Wallis ANOVA analysis did not yield a significant result, post hoc comparisons of the data obtained for specific pairs of experimental groups were not carried out.

### 3.3. Stopping at the Experimental Object

Stopping at the experimental object was relatively infrequent and was never observed during the tests with water. It was recorded only during one test with ethanol and four tests with acetic acid (Figure 2), in both cases at a rate of one observation per test. The overall analysis of the rate of occurrence of that behaviour carried out with the use of 2 × 3 Fisher Exact Probability Test discovered only a non-significant trend (Figure 2). As that test did not yield significant results, pairwise post hoc comparisons of the experimental groups by means of the 2 × 2 Fisher Exact Probability Test were not carried out.

The analysis of the latencies to the first episode of stopping close to the experimental object recorded during the tests with different compounds also discovered only a non-significant trend (Kruskal–Wallis ANOVA: *p* = 0.06). This result closely mirrors the result of the analysis of the rate of occurrence of the tests during which that behaviour was observed, and is related to the same phenomenon, absence of that behaviour during the tests with water, and its presence during a few tests with ethanol and acetic acid.

Kruskal–Wallis ANOVA analysis carried out to compare the numbers of observations of stopping close to the experimental object recorded during the tests with different compounds did not discover significant inter-group differences, or even a non-significant trend.

### 3.4. Withdrawal from the Experimental Object

Withdrawal from the experimental object was also relatively infrequent (Figure 3). However, this time Fisher Exact Probability Tests (the overall 2 × 3 one and post hoc 2 × 2 ones) discovered significant inter-group differences. The highest rate of withdrawal from the experimental object (23.3%) was recorded in the ant group tested with acetic acid (Figure 3). That result was significantly different from the ones obtained for the tests with water (no withdrawal observed) and with ethanol (one case of withdrawal observed during a single test) (Figure 3). Moreover, in the case of the tests with acetic acid, withdrawal from the experimental object was recorded not only at a rate of one observation per test (in the case of four tests), but also at a rate of two observations per test (in the case of further three tests).

The analyses of the latencies to the first episode of withdrawal from the experimental object and of the numbers of observations of that behaviour made during each of the tests also revealed statistically significant differences between the experimental groups (Kruskal–Wallis ANOVA: *p* < 0.01 in both cases). However, in the case of both these variables post hoc dyadic comparisons of the data obtained for specific pairs of experimental groups did not yield significant differences. Moreover, as that behaviour was observed at a maximum rate of two episodes per test, the results of the analysis of its latencies and of the numbers of its episodes observed during each test simply mirrored the results of the earlier analysis of the rate of occurrence of the tests during which it was recorded.

### 3.5. Bypassing the Experimental Object

Bypassing the experimental object (giving it a wide berth) was observed during the majority of the tests with acetic acid (83.3%), but never during the tests with water and with ethanol (Figure 4). The overall analysis of the data obtained for all tested groups yielded a highly significant result (Figure 4). Pairwise post hoc comparisons also revealed highly significant differences between the ants tested with acetic acid and each of the other two compounds (Figure 4).

Latencies to the first observation of bypassing of the experimental object also showed highly significant inter-group differences (overall Kruskal–Wallis ANOVA: *p* < 0.00001). Dyadic Siegel–Castellan post hoc tests also discovered highly significant differences (*p* < 0.00001) between the ants tested with acetic acid and either water or ethanol. However, this time, too, the results of the analysis of that variable also simply mirrored the results of the analysis of the rate of occurrence of the tests during which the tested ants engaged in that behaviour, and they did not document any new phenomenon.

The analysis of the numbers of cases of bypassing of the experimental object recorded during each test (overall Kruskal–Wallis ANOVA followed by pairwise post hoc tests) also discovered highly significant inter-group differences (Figure 5). However, this time the result of that analysis was related not only to presence or absence of bypassing of the experimental object during the tests with different compounds, but also to repeated expression of that behaviour during the tests with the acetic acid (up to three cases per test, with the median and both quartiles equal to 2; see Figure 5).

### 3.6. Jumping Away from the Experimental Object

Jumping away from the experimental object was absent in the group tested with water and relatively infrequent in the other two experimental groups (Figure 6). The overall analysis of these data by means of the 2 × 3 Fisher Exact Probability Test discovered only a non-significant trend (Figure 6) and was not followed by dyadic post hoc tests.

The analyses of the latencies to the first episode of that behaviour and of the numbers of cases of that behaviour observed during each test also discovered only non-significant trends (Kruskal–Wallis ANOVA: *p* = 0.07 in both cases) and, therefore, were not followed by dyadic post hoc tests. The results of all these analyses were related above all to the fact that jumping away from the experimental object was very infrequent (was observed at a rate of maximum one episode per test during the tests with ethanol, and maximum three episodes per test during the tests with acetic acid).

### 3.7. Antennal Contacts with the Experimental Object

Antennal contacts with the experimental object were observed during all the tests. In other words, the rate of occurrence of the tests during which that behaviour was observed was equal to one in the case of all experimental groups, and, consequently, did not show inter-group differences.

Kruskal–Wallis ANOVA analysis of the latencies to the first observation of antennal contacts with the experimental object did not discover any statistically significant differences between the compared groups, nor even a non-significant trend.

In contrast, Kruskal–Wallis ANOVA analysis of the numbers of observations of antennal contacts with the experimental object recorded during each test yielded a significant result (Figure 7). The lowest values of that variable were recorded in the case of the ants tested with water, and the highest ones in the group tested with ethanol (Figure 7). The difference between these two groups proved to be significant (Figure 7).

### 3.8. Nibbling of the Experimental Object (Not Accompanied or Accompanied by Gaster Flexing) and Nibbling of the Substrate in Its Vicinity

Nibbling of the experimental object not accompanied by gaster flexing, henceforth denoted in a simpler way as nibbling of the experimental object, was observed relatively frequently, on 19 out of 30 tests with water (63.3%), 25 out of 30 tests with ethanol (83.3%), and 22 out of 30 tests with acetic acid (73.3%) (Figure 8).

The overall analysis of these data carried out with the use of 2 × 3 Fisher Exact Probability Test discovered only a non-significant trend (Figure 8). As that test did not yield significant results, pairwise post hoc comparisons of the experimental groups by means of the 2 × 2 Fisher Exact Probability Test were not performed.

Kruskal–Wallis ANOVA analyses of the latencies to the first episode of nibbling of the experimental object and of the numbers of observations of that behaviour made during each of the tests did not reveal any statistically significant differences between the experimental groups, nor even any non-significant trends, and, therefore, they were not followed by dyadic post hoc tests.

The same results (absence of any significant differences or non-significant trends) were also obtained in two additional Kruskal–Wallis ANOVA analyses of the latencies to the first episode of nibbling of the experimental object and the numbers of observations of that behaviour made during each of the tests in which we took into account only the tests during which the ants engaged in that behaviour.

Two other subcategories of nibbling behaviour included in the list of behaviour patterns analysed in this study, nibbling of the experimental cotton pad accompanied by gaster flexing, and nibbling directed at the substrate close to the experimental object, were observed extremely infrequently. The first one was observed only once during a test with acetic acid. The second one was observed only three times during two tests with ethanol and during one test with acetic acid, always at a rate of one observation per test.

### 3.9. Self-grooming Behaviour Displayed Close to the Experimental Object

Self-grooming of the antennae close to the experimental object was observed much less frequently close to the cotton pad soaked in water [on 6 out of 30 tests (20.0%)] than close to the cotton pads soaked in the aqueous solution of ethanol [on 24 out of 30 tests (80.0%)] and acetic acid [on 17 out of 30 tests (56.7%)] (Figure 9). In other words, the rate of occurrence of the tests during which the ants performed that behaviour was exactly four times higher in the case of ethanol than in the case of water. The overall analysis of the data obtained for all experimental groups carried out by means of the 2 × 3 Fisher Exact Probability Test discovered highly significant inter-group differences (Figure 9). Post hoc pairwise comparisons made with the use of 2 × 2 Fisher Exact Probability Test also revealed a highly significant difference in the case of the comparison water–ethanol, and a slightly less significant one in the case of the comparison water–acetic acid. A non-significant trend was also discovered in the case of the comparison ethanol–acetic acid (Figure 9).

Closely similar results were also yielded by the analysis of the latencies from the start of each test to the first occurrence of self-grooming of the antennae close to the experimental object (Figure 10). Kruskal–Wallis ANOVA discovered highly significant differences between the compared groups, and subsequent pairwise Siegel–Castellan post hoc tests also revealed significant differences in the case of the comparisons water–ethanol and water–acetic acid, and a non-significant trend in the case of the comparison ethanol–acetic acid (Figure 10). The analysed variable took the highest values in the case of the ants tested with water and the lowest values in the case of the ants tested with ethanol.

Kruskal–Wallis ANOVA analysis of the numbers of bouts of self-grooming of the antennae close to the experimental object also discovered highly significant differences between the compared groups, and subsequent pairwise Siegel–Castellan posthoc tests revealed further significant differences in the case of the comparisons water–ethanol and water–acetic acid (Figure 11). However, this time the comparison ethanol–acetic acid did not discover a non-significant trend (Figure 11). The analysed variable took the highest values in the case of the ants tested with ethanol, and the lowest values in the case of the ants tested with water.

As can be seen, statistical analysis of three variables characterizing self-grooming of the antennae taking place close to the experimental object brought about closely similar results. Without further analysis, it was, however, impossible to determine to what degree the effects detected by the analysis of the latencies to the first occurrence of that behaviour during each test (Figure 10), and the numbers of bouts of that behaviour observed during each test (Figure 11) did simply mirror the differences in the rate of occurrence of the tests during which it was observed (Figure 9), as they might also have been at least partly related to behavioural differences between the ants that did actually engage in self-grooming of the antennae close to the experimental object. To throw more light on that question, in two additional analyses, we took into account only the tests during which that behaviour was actually observed (Figure 12 and Figure 13).

This time, too, the shortest latencies to the first occurrence of the analysed behaviour were observed in the case of the ants tested with ethanol, and the longest ones in the case of the ants tested with water (Figure 12). However, inter-group differences were now no longer statistically significant. The overall analysis of these data by means of Kruskal–Wallis ANOVA discovered only a non-significant trend (Figure 12).

In contrast, Kruskal–Wallis ANOVA analysis of the numbers of bouts of self-grooming of the antennae close to the experimental object recorded during each test yielded a significant result also when the tests during which that behaviour was not observed were discarded from analysis (Figure 13). Similarly as in the case of the analysis in which all the tests were taken into account, the lowest values of that variable were recorded in the case of the ants tested with water, and the highest ones in case of the ants tested with ethanol (Figure 13). However, this time pairwise post hoc tests carried out to compare the differences between the ants tested with water and with two experimental compounds (ethanol and acetic acid) discovered only non-significant trends (Figure 13).

Self-grooming of the gaster carried out close to the experimental object was much less frequent than self-grooming of the antennae. It was observed only once during one test with water and in total ten times during four tests with strongly smelling experimental compounds: once during two tests with acetic acid, twice during one test with ethanol and one test with acetic acid, and four times during yet another test with acetic acid. Statistical analyses of three variables quantifying that behaviour (rate of occurrence of the tests during which it was observed, latency to its first appearance during each test, and number of observations of that behaviour during each test) carried out by means of 2 × 3 Fisher Exact Probability Test and Kruskal–Wallis ANOVA did not discover any significant inter-group differences or even non-significant trends.

### 3.10. Aggressive Behaviour Directed at a Nestmate Close to the Experimental Object

The present experiment also yielded data documenting the effect of experimental compounds on aggressive social behaviour of the tested ants. Thus, charges at a nestmate present close to the experimental object were never observed during the tests with water, but were observed during the tests with both ethanol and acetic acid (16.7% and 26.7% of the tests, respectively) (Figure 14). The overall analysis of the data obtained for all experimental groups carried out by means of the 2 × 3 Fisher Exact Probability Test discovered significant inter-group differences (Figure 14). Post hoc pairwise comparisons made with the use of 2 × 2 Fisher Exact Probability Test also revealed significant differences in the case of the comparisons water–ethanol and water–acetic acid. However, the results obtained for ethanol and acetic acid did not differ significantly (Figure 14).

The analyses of the latencies to the first charge at a nestmate observed close to the experimental object and of the numbers of observations of that behaviour recorded during each test also yielded significant differences (Kruskal–Wallis ANOVA: *p* = 0.01 in both cases). However, pairwise post hoc tests did not discover any significant results or non-significant trends.

In two additional analyses (Mann–Whitney U test), we also compared the values of these two variables, taking into account only the tests during which the ants engaged in the analysed behaviour. As charges at a nestmate were never recorded in the case of the ants tested with water, only the results obtained for the ants tested with ethanol and acetic acid were compared. None of these analyses yielded a significant result or discovered a non-significant trend.

Two other subcategories of aggressive behaviour directed to nestmates, open-mandible threat and dragging of a nestmate, were observed even less frequently than charges at a nestmate. They also were never observed close to the experimental cotton pads soaked in water. Open-mandible threat directed to a nestmate ant present close to the experimental object was recorded only six times during three tests with ethanol and three tests with acetic acid, in all cases at a rate of one observation per test. Dragging of a nestmate close to the experimental object was observed only twice, once during a test with ethanol, and once during a test with acetic acid. Statistical analyses of three variables quantifying these subcategories of aggressive behaviour carried out by means of 2 × 3 Fisher Exact Probability Test and Kruskal–Wallis ANOVA did not discover any significant inter-group differences or even any non-significant trends.

### 3.11. Friendly Social Behaviour Directed at a Nestmate Close to the Experimental Object

Friendly antennal contacts with a nestmate taking place close to the experimental object were also observed relatively infrequently. In total only fourteen observations of that behaviour were recorded during the whole experiment. That behaviour was recorded at a rate of one observation per test during three tests with water, three tests with ethanol and one test with acetic acid, two observations per test during one test with ethanol and two tests with acetic acid, and three observations per test during yet another test with acetic acid. Only three cases of that behaviour were recorded during the tests with water; the remaining eleven observations were all recorded during the tests with ethanol and acetic acid. However, statistical analyses of three variables quantifying that behaviour carried out by means of 2 × 3 Fisher Exact Probability Test and Kruskal–Wallis ANOVA did not discover any significant results or non-significant trends.

Lastly, allogrooming of a nestmate close to the experimental object was observed only once, during a test with acetic acid.

## 4. Discussion

### 4.1. The Effects of Exposure to Water, Ethanol and Acetic Acid on Behaviour of the Tested Ants

The present study yielded clear and coherent results documenting a wide spectrum of behavioural effects of exposure of ants (extranidal workers of the narrow-headed ant *Formica exsecta*) to water and aqueous solutions of ethanol and acetic acid presented to them close to their field nests. The responses of the ants to these three categories of experimental objects showed a number of significant and in some cases even highly significant differences. Significant modifications of ant behaviour related to exposure to ethanol and/or acetic acid (*p* ≤ 0.05) were observed in the case of the variables quantifying five behavioural categories: withdrawal from the experimental object, bypassing of that object, antennal contacts with that object, self-grooming of the antennae close to that object, and charge at a nestmate close to that object. Furthermore, non-significant trends (0.05 < *p* ≤ 0.1) were also discovered in the case of the variables quantifying further four behavioural categories: ignoring the experimental object, stopping at that object, jumping away from that object, and nibbling of that object. It also should be noted that 10 out of 16 behavioural categories recorded during this experiment (stopping close to the experimental object, withdrawal from that object, bypassing of that object, jumping away from that object, nibbling of that object accompanied by gaster flexing, nibbling of the substrate close to that object, charge at a nestmate close to that object, open-mandible threat directed at a nestmate close to that object, dragging of a nestmate close to that object and allogrooming of a nestmate close to that object) were observed only during the tests with ethanol and/or acetic acid, but were never observed during the tests with water.

We also would like to point out that absence of significant inter-group differences was observed in the case of behavioural categories that were recorded both very infrequently (for instance, nibbling accompanied by gaster flexing, or dragging of a nestmate close to the experimental object), and very frequently, in some cases even on all the tests, such as ignoring of the experimental object and antennal contacts with it.

### 4.2. Differences between the Effects of Ethanol and Acetic Acid

Our findings clearly demonstrated that acetic acid was aversive for the ants. In particular, acetic acid was found to induce withdrawal from the cotton pad soaked in its solution, jumping away from that object and, most importantly, very frequent bypassing of that object (giving it a wide berth). In contrast, ethanol induced spontaneous interest and attraction of the ants and exerted a stimulatory effect on their exploratory behaviour, enhancing antennal exploration of the cotton pads soaked in its solution and self-grooming of the antennae taking place close to such objects. We did not check if exposure to ethanol increases general exploratory tendencies of the ants (e.g., makes them more likely to antennate a different novel object). However, we checked if exposure to ethanol influences the readiness of the ants to show some forms of exploratory behaviour other than antennation of the test object, namely, antennation of a nestmate ant present in its vicinity, and three various subcategories of nibbling behaviour. In the ants, nibbling may also act as exploratory behaviour, in contrast to true biting [70]. However, no significant inter-group differences were discovered by these analyses. In particular, nibbling of the experimental cotton pad was observed relatively frequently in all three groups, also in response to the cotton pad soaked in water. The analysis of inter-group differences in the rate of occurrence of the tests during which that behaviour was observed discovered only a non-significant trend.

Different modifications of some behaviour patterns observed close to the cotton pads soaked in either ethanol or acetic acid imply that at least some of the effects of exposure of the tested ants to these two compounds cannot be reduced to general effects of exposure to strong olfactory stimulation. The responses of the ants to ethanol and acetic acid showed particularly important differences in the case of two behavioural categories documenting aversiveness of acetic acid to the ants: withdrawal from the experimental object and bypassing of the experimental object. In both these cases, significant differences between the behaviour of ants responding to these two compounds were detected by the analyses of the rate of occurrence of the tests during which that behaviour was expressed, and in the case of bypassing of the experimental object also by the analyses of the latencies to the first observation of that behaviour, and the numbers of observations of that behaviour recorded during each test.

The ants also tended to engage more frequently in self-grooming of the antennae close to the cotton pads soaked in ethanol than close to identical cotton pads soaked in acetic acid. However, in the case of this behavioural category comparison of the results obtained for the ants exposed to ethanol and to acetic acid discovered only non-significant trends (in the case of the rate of occurrence of the tests during which that behaviour was observed, and of the latencies to the first episode of that behaviour during each test).

It is also worth stressing that both experimental compounds used in our study influenced not only locomotion of the tested ants (including escape behaviour), their exploratory behaviour and self-grooming behaviour, but also their aggressive social behaviour. In particular, charges at a nestmate ant were significantly more frequent close to the cotton pads soaked in either ethanol or acetic acid than close to the cotton pads soaked in water. As already pointed out in the Introduction, similar manifestations of ethanol-induced aggression were also documented in the honeybees [24,30] and in the fruit flies [20]. However, interrelationships between insect aggressive behaviour and effects of ethanol are not limited to that phenomenon, as in other studies ethanol consumption did not influence stinging behaviour of the tested honeybee workers [21] or was even followed by a decreased level of defensive behaviour of the tested bees, a phenomenon related most probably to analgesic effects of ethanol consumption [31]. It also should be noted that in our present experiment increased rate of occurrence of charges at nestmate ants was observed not only in response to ethanol, but also in response to acetic acid. Therefore, in our present study this modification of ant aggressive behaviour was not induced in a specific manner solely by ethanol. In other words, a possible explanation of that finding may lie in the fact that some behaviour patterns observed in the present study might have been triggered by strong olfactory stimulation acting in the same way irrespectively of the specific compound to which the ants were exposed.

### 4.3. The Effects of Exposure to Different Experimental Compounds on Behaviour Patterns Fulfilling Similar Functions

It is also worth noting that behavioural variables quantifying behaviour patterns fulfilling similar functions depended on experimental factors in a similar way even if inter-group differences were not significant. This rule is well illustrated by the results of the analysis of three subcategories of aggressive behaviour observed during our tests: charges at a nestmate, open-mandible threats directed to a nestmate, and dragging of a nestmate. The differences between experimental groups proved to be significant only in the case of charges at a nestmate. However, all three discussed patterns of aggressive behaviour were never observed close to the cotton pads soaked in water, and all of them were observed close to the cotton pads soaked either in ethanol or in acetic acid, even if they were recorded very infrequently.

Similarly, aversiveness of acetic acid for the ants was also illustrated by the results of the analysis of three variables, withdrawal from the experimental object, bypassing of that object, and jumping away from it. However, differences between experimental groups treated with acetic acid and with other experimental compounds were significant only in the case of the first two behavioural categories. In the case of the rate of the occurrence of the third behavioural category fulfilling a similar function only a non-significant trend was discovered.

### 4.4. Ethanol in Ant Exocrine Glands and Its Role in the Mediation of Ant Behaviour

So far, our present knowledge concerning the responses of the ants to ethanol and acetic acid and the role of these compounds in the mediation of ant behaviour and physiology was far less complete and advanced than a huge body of experimental findings documenting behavioural effects of consumption of/exposure to ethanol in two insect species most frequently studied in the research devoted to that question, the fruit fly (*Drosophila melanogaster*) and the honeybee (*Apis mellifera*). The first data on alcohol intoxication experimentally induced in ant workers and on responses of ants to intoxicated nestmates and non-nestmates were provided already in the classical book of Sir John Lubbock (1884) [71]. Since a relatively long time ago, it has also been known that both ethanol and acetic acid can be found in various exocrine glands of various ant species. In particular, ethanol was found in the Dufour’s glands and in the mandibular glands of several species of ants from the genus *Myrmica* [72,73,74,75], and in the Dufour’s glands of workers of another myrmicine ant species, *Manica rubida* [76]. In workers of *Myrmica rubra,* ethanol was also found to increase sinuosity of locomotion, to act as an attractant in synergy with the butanone, and to induce the deposition of the Dufour’s gland secretions in the foraging area [72]. Attractiveness of ethanol has also been documented in other solitary and social insects, such as *Drosophila* fruit flies (adults and larvae) [19,53] and honeybees [23,32].

### 4.5. Acetic Acid in Ant Exocrine Glands and Its Role in the Mediation of Ant Behaviour

Acetic acid was found to be present, among others, in the metapleural glands of the leafcutter ant *Acromyrmex octospinosus* [77,78]. Some literature data illustrating aversiveness of the acetic acid for the ants have been reported already more than a century ago. Thus, in one of the first studies devoted to ant learning [79] ants from an unspecified species were trained to escape from a vinegar-saturated platform by jumping on a vinegar-free substrate. Acetic acid was also found to be both toxic and repellent to the ants of the species *Lasius niger* and *Crematogaster matsumurai*. In a laboratory experiment, all workers of these two ant species were killed within 10–60 min after the start of their exposure to acetic acid. In a subsequent field experiment, no ant was observed to approach the tray with sucrose on which a small (1 µL) drop of acetic acid has been deposited. Out of 17 tested compounds such strong (100%) repellency for the ants was observed only in the case of acetic acid [80].

Acetic acid was also found to induce alarm behaviour of workers of *Lasius niger*. As soon as it was applied to the tray containing sucrose, most of the swarming ants started to disperse, and some were observed to engage in gaster flexing (a form of threatening behaviour), and to attack each other. After a few minutes this alarm response was followed by recruitment of other workers. However, workers of *Crematogaster matsumurai* did not display such alarm behaviour and responded to acetic acid only by random movements and evasion [80]. Interestingly, in another species from that genus, *Crematogaster scutellaris*, acetic acid was found to act as the principal alarm pheromone. Workers of that species showed excitement and dispersed when a bit of filter paper impregnated with acetic acid was placed on a platform 1 cm above the place at which they were collectively feeding [81].

Vinegar is also commonly recommended by numerous internet sites as one of the most efficient natural ant repellents that may help us get rid of the ants in case of their undesirable appearance in our houses and gardens [82,83].

### 4.6. Limitations of the Present Work and Perspectives of Future Research

The most important limitations of our present work are related to the fact that our field methods did not allow us to monitor the behaviour of individual ants. In the future field studies investigating similar issues the use of individually marked ant workers might bring about additional insights concerning the causal factors underlying the investigated behavioural phenomena.

Nevertheless, we would like to stress that, in contrast to laboratory experiments, field experiments such as the one carried out in our present study may provide important methodological advantages. In particular, field experiments make possible the expression of behaviour patterns allowing the tested ants to escape from the source of the tested compound(s). Consequently, some behavioural effects of exposure to acetic acid that were absent or very weak in our earlier laboratory tests in which the ants could not escape from exposure to that compound were well expressed in our field experiment. Our present findings imply thus that field experiments may act as a useful tool in the research on behavioural effects of ethanol and other neuroactive compounds observed in various insect species, and, therefore, their contribution to that research should be highly recommended.

Comparative studies using as subjects ants from various species should also be highly recommended for future research. As already shown by the previous research, presence/absence of ethanol in exocrine glands of the ants of the genus *Myrmica* varies depending on the species [72,73,74,75,76]. Similarly, in a field experiment, workers of the ant species *Lasius niger* and *Crematogaster matsumurai* were found to respond differently to acetic acid [80], and the role of acetic acid in the induction of alarm behaviour proved to be strikingly different in two species from the same ant genus *Crematogaster* [80,81]. Lastly, comparative research on the effects of phenotypic plasticity of ant workers on the investigated behavioural phenomena may also bring about interesting results, as earlier studies on ethanol responses of both solitary and social insects already documented such effects in the case of both *Drosophila* fruit flies [42] and honeybees [45,46].

Future experimental studies may also elucidate some unsolved questions raised by the present study. For instance, there could be various reasons why the ants ignored the experimental object or stopped at it but did not engage in any further physical contacts with it, and we cannot exclude that at least in some cases the ants failed to perceive the test object. The behavioural categories of “ignoring the experimental object” and “stopping at the experimental object” might also have been heterogenous and might have included both the cases in which the ants simply did not perceive the experimental object and the cases in which they perceived it but for some reason did not engage in further behavioural responding to it. At present we have no means to unequivocally solve that question on the basis of our behavioural observations. We may only note that if the tested ants were indeed unaware of the presence of an experimental object, we might rather expect that, at least in a part of cases, we will observe their headlong bouncing into that object, and such behaviour was not observed. However, future research will perhaps be able to shed more light on that question. If the behaviour of the ants would be videorecorded, the analysis of these recordings might discover some subtle indications that the ants perceived the ignored object, such as, for instance, augmented rate of antennation observed close to that object, but not followed by antennal contacts with it, or orienting the head toward that object when passing by.

Similarly, in the future research, we could not only record the occurrence of particular behaviour patterns, but also analyse the sequences of behavioural responses. For instance, we might find out that increased self-grooming of the antennae observed close to the cotton pads soaked with ethanol might be followed by modifications of responses to nestmates, as self-grooming of antennae may increase sensory acuity of olfactory perception of the tested ants.

Behavioural effects of administration of/exposure to two compounds investigated in this study, ethanol and acetic acid, have yet another interesting aspect that also could be investigated in the future, namely, concentration-dependent modulation of behaviour. Previous research on the effects of ethanol on behaviour of insects from two model species most frequently investigated in that context, the fruit flies *(Drosophila melanogaster*) and the honeybees (*Apis mellifera*), often involved the administration of ethanol in various concentrations, and the concentration of the ethanol solution had usually crucial importance for the obtained results (*Drosophila* flies: [11,53]; honeybees: [21,23,26,27,28,29,30,32,33]). Concentration-dependent modulation of behaviour was usually investigated in the experiments during which ethanol was administered orally (*Drosophila* flies: [11,53]; honeybees: [21,23,26,27,28,29,30,32,33]), but the effects of different concentrations of ethanol were also checked in some studies that involved exposure of the tested insects to vapours of that compound (*Drosophila* flies: [39,40]). The future experiments during which the ants would be exposed to cotton pads soaked in aqueous solutions of ethanol and acetic acid administered at various concentrations might, for instance, shed light on the question if acetic acid will remain to be so strongly aversive for the ants also when it will be administered at a lower concentration than the one used in the present study.

As can be seen, both the results of our study and the perspectives of future research suggest that ants might be used as yet another model animal species in the future research on biological roots and correlates of alcoholism.

## 5. Conclusions

Our present field experiment belongs to a small number of studies carried out to investigate the role of two important compounds, ethanol and acetic acid, in the mediation of a wide spectrum of ant behaviour patterns (in total 16 behavioural categories). The results of our experiment confirmed that both these compounds can induce significant modifications of ant behaviour influencing various aspects of locomotion, exploratory behaviour, self-grooming behaviour, and aggressive social behaviour. Our findings also confirmed that whereas acetic acid is clearly aversive for the ants, ethanol induces their interest and enhances antennal exploration of its source and subsequent self-grooming of the antennae. Both ethanol and acetic acid were also found to enhance relatively mild forms of aggressive behaviour directed by the tested ant workers to their nestmates. It is worth stressing that these findings have been obtained in a field experiment in which the tested ant workers had a possibility to escape from the contact with the experimental compounds. Therefore, these findings are more closely related to behavioural processes taking place in the natural environment of the tested ants than the results of laboratory experiments. Contribution of field experiments to the research on behavioural effects of ethanol and other neuroactive compounds in various insect species should thus be strongly recommended. To conclude, our present results documented a wide spectrum of behavioural effects of exposure to ethanol and acetic acid in a highly social animal species tested in the conditions of the natural environment and contributed to the broadening of our general knowledge about behavioural responses to these two compounds encountered in the animal world. Both the results of our study and the perspectives of future research also suggest that ants might be used as yet another model animal species in the future research on biological roots and correlates of alcoholism.

## Figures and Tables

**Figure 1 animals-13-02734-f001:**
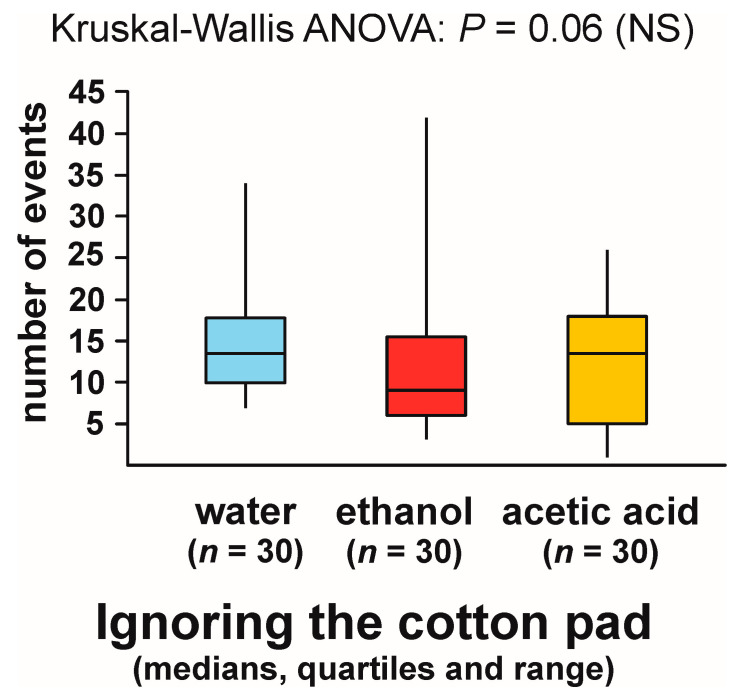
The numbers of observations of ignoring the experimental object (a cotton pad soaked in water, aqueous solution of ethanol, or aqueous solution of acetic acid) recorded during field tests with workers of the narrow-headed ant *Formica exsecta*. NS: not significant.

**Figure 2 animals-13-02734-f002:**
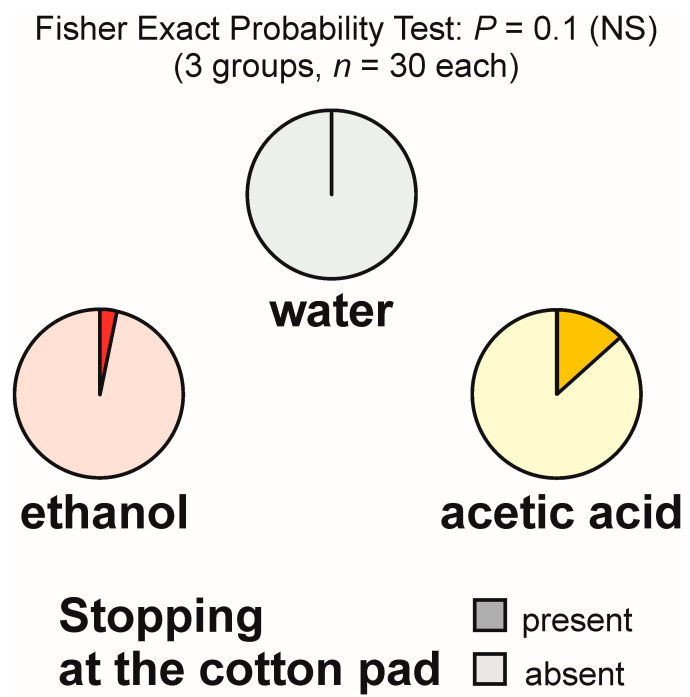
The rate of occurrence of the tests during which at least one worker of the narrow-headed ant (*Formica exsecta*) was observed to stop close to the experimental object (a cotton pad soaked in water, aqueous solution of ethanol, or aqueous solution of acetic acid). The data obtained for all tested groups were compared by means of the 2 × 3 Fisher Exact Probability Test.

**Figure 3 animals-13-02734-f003:**
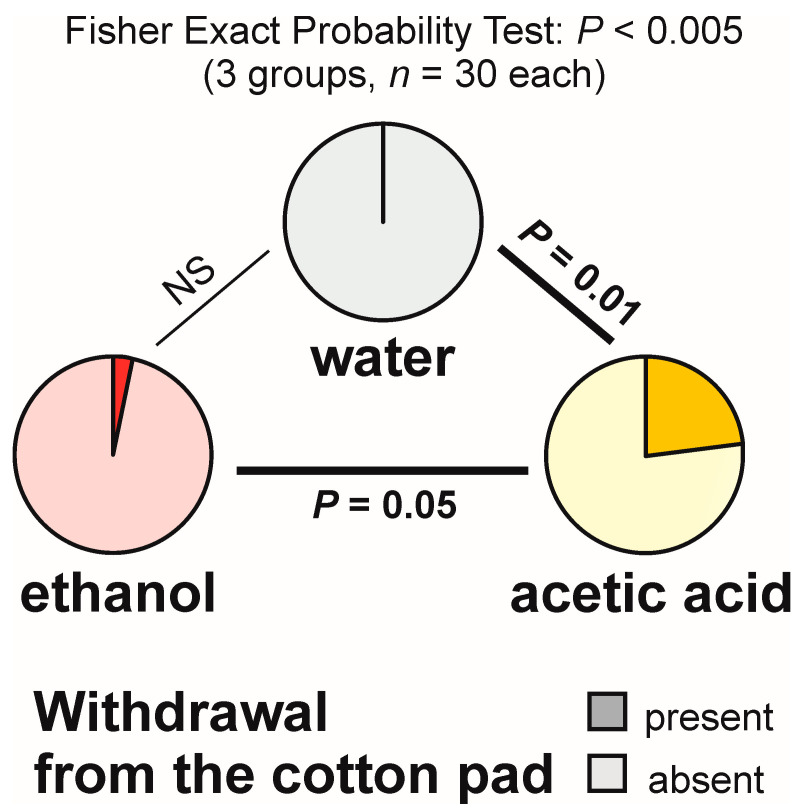
The rate of occurrence of the tests during which at least one worker of the narrow-headed ant (*Formica exsecta*) was observed to withdraw from the experimental object (a cotton pad soaked in water, aqueous solution of ethanol, or aqueous solution of acetic acid). The data obtained for all tested groups were compared by means of the 2 × 3 Fisher Exact Probability Test followed by pairwise post hoc comparisons carried out by means of the 2 × 2 Fisher Exact Probability Test.

**Figure 4 animals-13-02734-f004:**
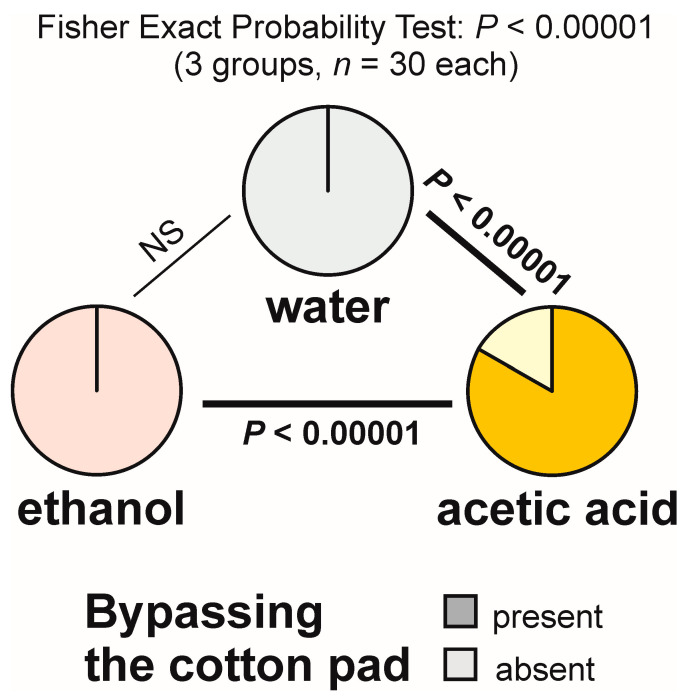
The rate of occurrence of the tests during which at least one worker of the narrow-headed ant (*Formica exsecta*) was observed to bypass the experimental object (a cotton pad soaked in water, aqueous solution of ethanol, or aqueous solution of acetic acid). The data obtained for all tested groups were compared by means of the 2 × 3 Fisher Exact Probability Test followed by pairwise post hoc comparisons carried out by means of the 2 × 2 Fisher Exact Probability Test.

**Figure 5 animals-13-02734-f005:**
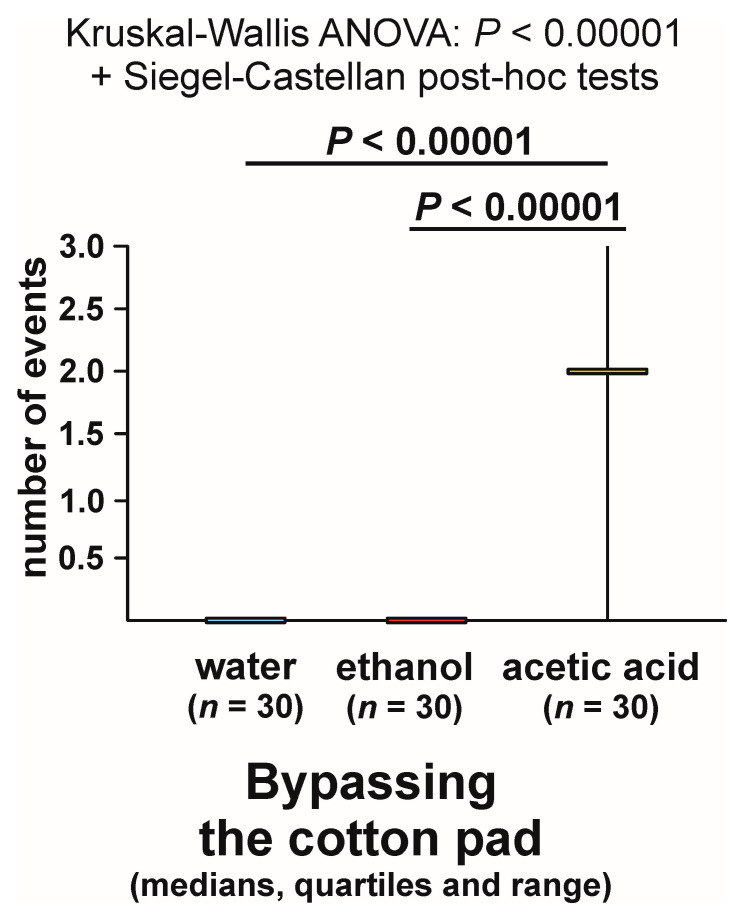
The numbers of cases of bypassing the experimental object (a cotton pad soaked in water, aqueous solution of ethanol, or aqueous solution of acetic acid) observed during the field tests with workers of the narrow-headed ant (*Formica exsecta)*.

**Figure 6 animals-13-02734-f006:**
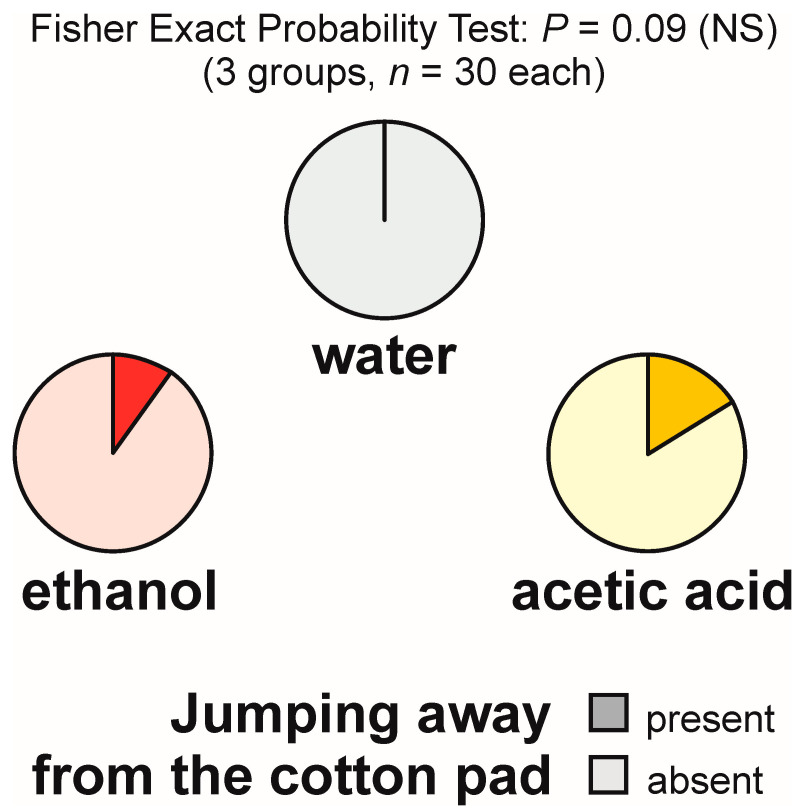
The rate of occurrence of the tests during which at least one worker of the narrow-headed ant (*Formica exsecta*) was observed to jump away from the experimental object (a cotton pad soaked in water, aqueous solution of ethanol, or aqueous solution of acetic acid). The data obtained for all tested groups were compared by means of the 2 × 3 Fisher Exact Probability Test.

**Figure 7 animals-13-02734-f007:**
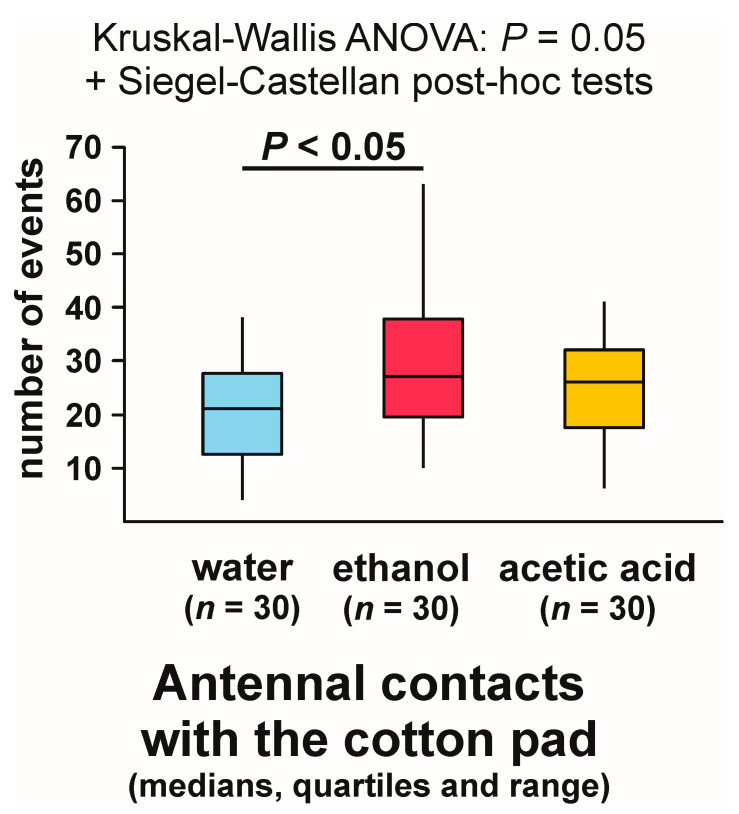
The numbers of observations of antennal contacts with the experimental object (a cotton pad soaked in water, aqueous solution of ethanol, or aqueous solution of acetic acid) observed during the field tests with workers of the narrow-headed ant (*Formica exsecta)*.

**Figure 8 animals-13-02734-f008:**
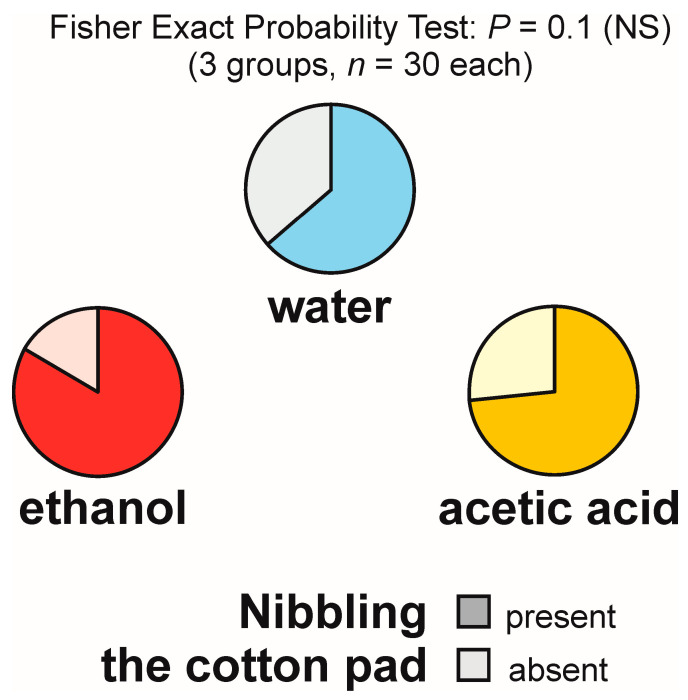
The rate of occurrence of the tests during which at least one worker of the narrow-headed ant (*Formica exsecta*) was observed to engage in nibbling of the experimental object (a cotton pad soaked in water, aqueous solution of ethanol, or aqueous solution of acetic acid). The data obtained for all tested groups were compared by means of the 2 × 3 Fisher Exact Probability Test.

**Figure 9 animals-13-02734-f009:**
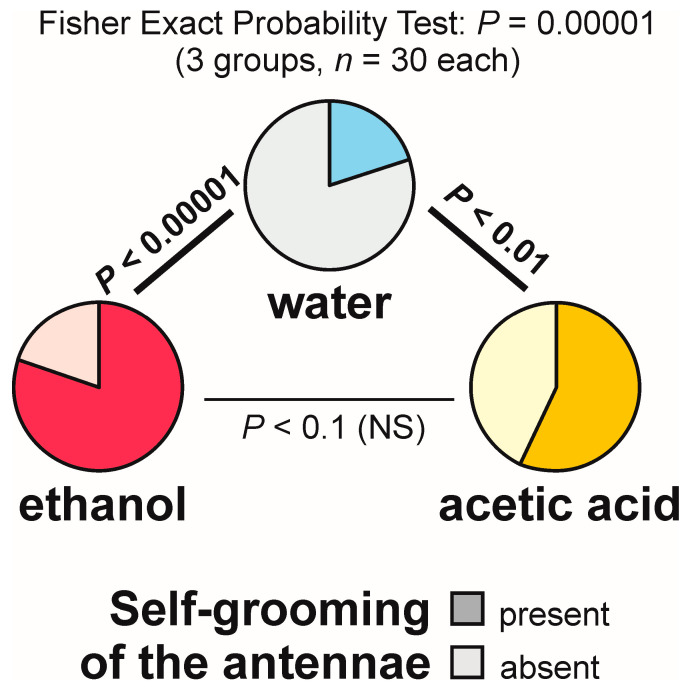
The rate of occurrence of the tests during which at least one worker of the narrow-headed ant (*Formica exsecta*) was observed to engage in self-grooming of the antennae close to the experimental object (a cotton pad soaked in water, aqueous solution of ethanol, or aqueous solution of acetic acid). The data obtained for all tested groups were compared by means of the 2 × 3 Fisher Exact Probability Test followed by pairwise post hoc comparisons carried out by means of the 2 × 2 Fisher Exact Probability Test.

**Figure 10 animals-13-02734-f010:**
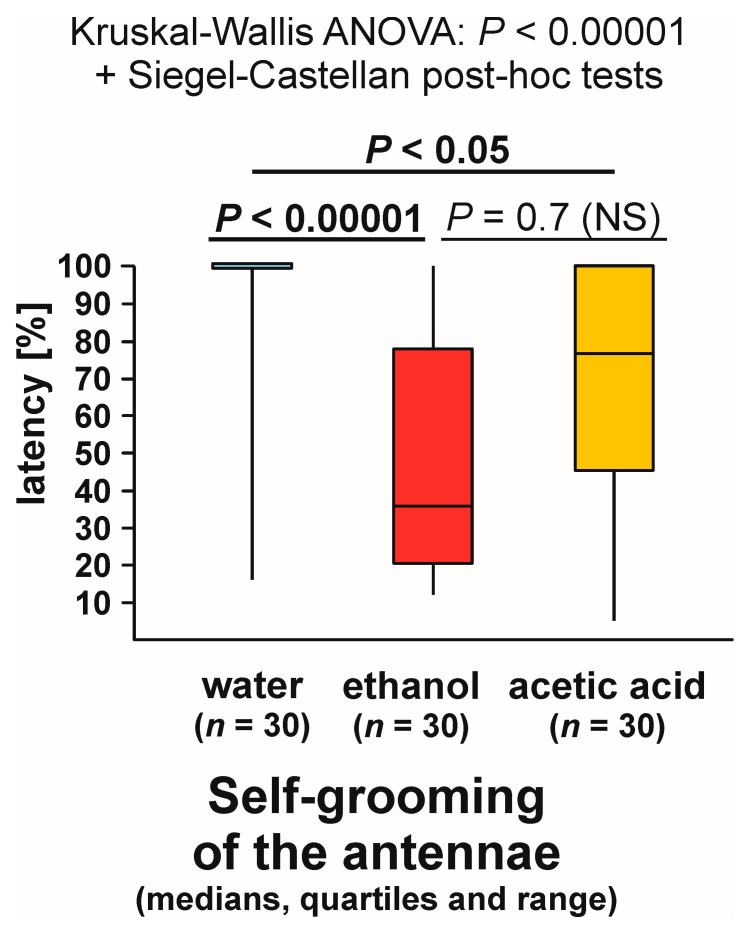
The latencies from the start of the test to the first episode of self-grooming of the antennae close to the experimental object (a cotton pad soaked in water, aqueous solution of ethanol, or aqueous solution of acetic acid) observed during the field tests with workers of the narrow-headed ant (*Formica exsecta*). Latencies are expressed as the per cent of the total test time.

**Figure 11 animals-13-02734-f011:**
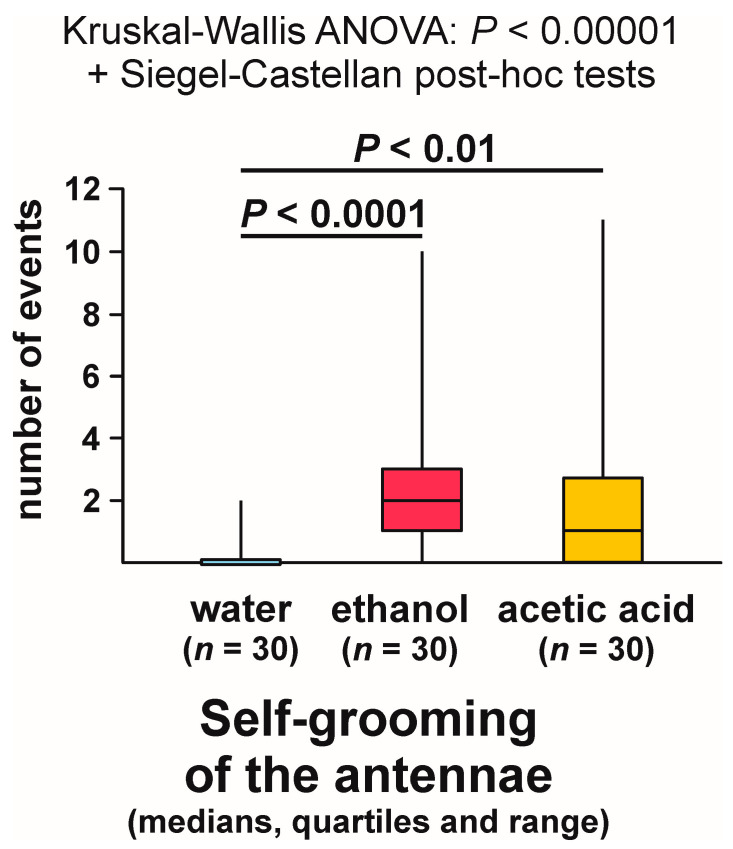
The numbers of bouts of self-grooming of the antennae close to the experimental object (a cotton pad soaked in water, aqueous solution of ethanol, or aqueous solution of acetic acid) observed during the field tests with workers of the narrow-headed ant (*Formica exsecta*).

**Figure 12 animals-13-02734-f012:**
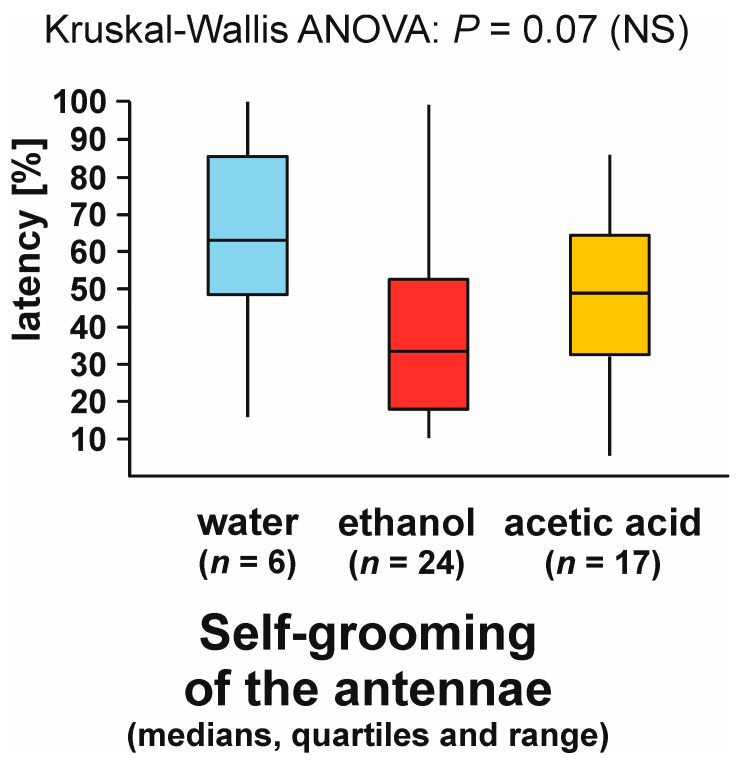
The latencies from the start of the test to the first episode of self-grooming of the antennae close to the experimental object (a cotton pad soaked in water, aqueous solution of ethanol, or aqueous solution of acetic acid) observed during the field tests with workers of the narrow-headed ant (*Formica exsecta*). The tests during which that behaviour was not observed were discarded from the analysis. Latencies are expressed as the per cent of the total test time.

**Figure 13 animals-13-02734-f013:**
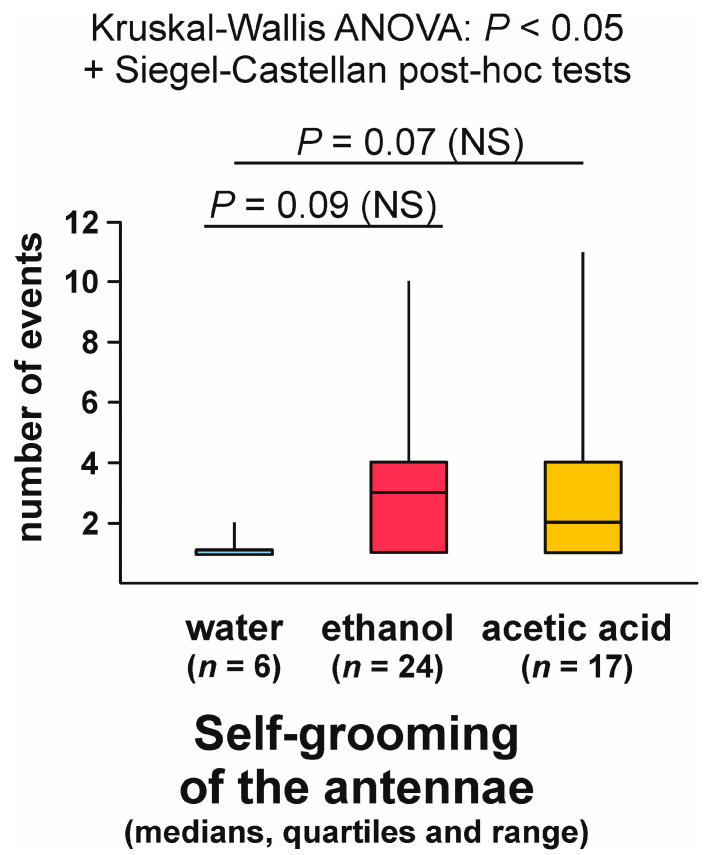
The numbers of bouts of self-grooming of the antennae close to the experimental object (a cotton pad soaked in water, aqueous solution of ethanol, or aqueous solution of acetic acid) made during the field tests with workers of the narrow-headed ant (*Formica exsecta)*. The tests during which that behaviour was not observed were discarded from the analysis.

**Figure 14 animals-13-02734-f014:**
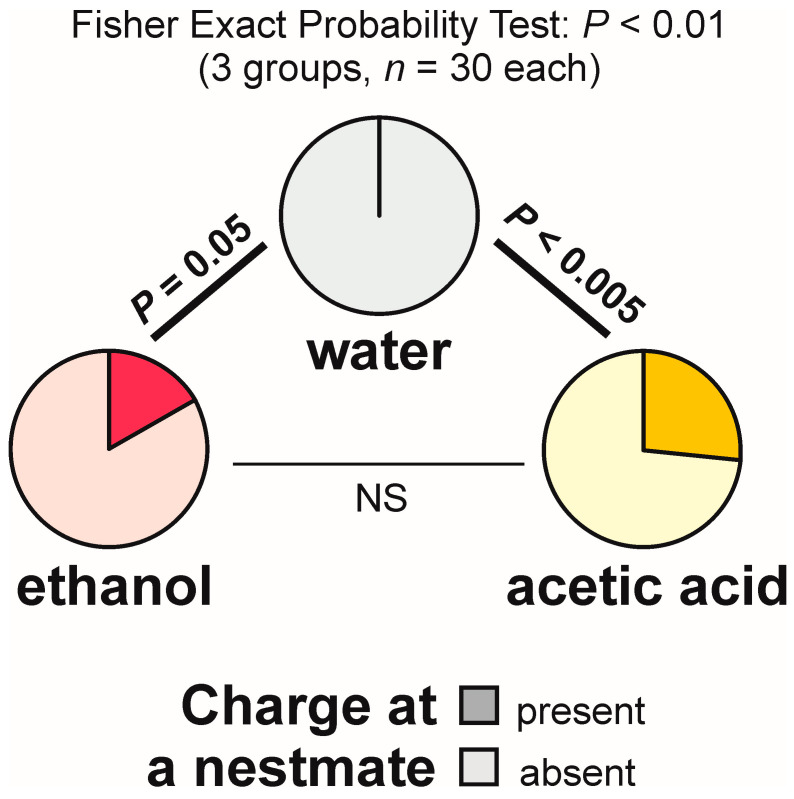
The rate of occurrence of the tests during which at least one worker of the narrow-headed ant (*Formica exsecta*) was observed to charge at a nestmate close to the experimental object (a cotton pad soaked in water, aqueous solution of ethanol, or aqueous solution of acetic acid). The data obtained for all tested groups were compared by means of the 2 × 3 Fisher Exact Probability Test followed by pairwise post hoc comparisons carried out by means of the 2 × 2 Fisher Exact Probability Test.

## Data Availability

Additional data can be found in keeping of the Laboratory of Ethology of the Nencki Institute of Experimental Biology, and they are accessible upon reasonable request from the corresponding author.

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
