# Peer review of "The Effects of Ethanol and Acetic acid on Behaviour of Extranidal Workers of the Narrow-Headed Ant Formica exsecta (Hymenoptera, Formicidae) during a Field Experiment"

_animals, 2023, doi:10.3390/ani13172734_

Round 1

Reviewer 1 Report

In this work, the effects of two compounds, ethanol and acetic acid, were investigated on the behaviour of the narrow-headed ant Formica exsecta in the field. This topic is interesting and represents a very intensive work that calls for respect, as at least 16 parameters related to ant behaviour in response to ethanol and acetic acid were studied by the authors.

However, the manuscript is very difficult to read and could lead to some confusions, as a clear "continuity" between each section is lacking. For example, the Discussion does not seem to bring a thorough reflexion about what the authors observed in this work, but to be a list of results with some citations of what is similar or different in other insect species. Thus, this manuscript requires considerable editing to be clearer and more understandable. There are listed my comments :

1. The Simple Summary is almost as long as the Abstract and does not only expose the main content of the manuscript : scientific question, experiment, results. That makes both sections quite similar and, consequently, repetitive.

2. In the Introduction, the goal, the interest and the global context of the study are not well defined. The authors provide the reader with a lot of examples of effects of ethanol and acetic acid on the behaviour/functions of several species of insects but donnot introduce the species on which they worked. Instead, they introduce F. exsecta in the Material & Methods section. The biology and sophisticated social behaviour (as said by the authors) of F. exsecta should be developed in the Introduction, as it is one of the reasons this species was chosen for the study. The effects of ethanol and acetic acid on other species should not be so developed, maybe they could be settled in "blocks" : one block of citations about the effects on a behaviour in particular, another block for another behaviour... without precising the species each time (it does not bring any relevant information in the Introduction, and if some point has to be developed, the species can be named in the Discussion without being too repetitive). The Introduction should end on the interest and main goal of the study instead of on a comparison of the advantages of field and laboratory experiments. These points can be developed in the Materials & Methods section, even in the Discussion.

3. In the Materials & Methods, the authors indicate that the experiments were videorecorded, that seems logical since 16 parameters/test were taken into account. However, they indicated it in the section 2.3. (dealing with the compounds which were used) and not in 2.4. (dealing with the tests which were performed). This can lead to confusion, the reader could think all these parameters were eye-observed simultaneously in the field, and then could give minor "credit" to the experimental design.

4. Curiosity : why 8 ants in the "pre-selection" circles before tests ? Is this a relevant number considering the size of such ant colony ? Does it reflect something important ?

5. Also in the Material & Methods, maybe the authors could write only one section about the species, indicating the colony's location, in the condition that they introduced F. exsecta in the Introduction (see 2.).

6. In the Results, section 3.1., the authors wrote "modifications of locomotion". What are they refering to ? A major concern is that the presentation of the results is very confusing and too repetitive, which tends to "lose" the reader. There are many figures reflecting only one result for one behaviour/parameter, while all the results could be resumed in three more synthetic tables : occurrence, number of observations, latency. From these tables, the most relevant observation could then be higlighted and discussed more easily.

7. In the Discussion, other cases of effects of ethanol and acetic acid are cited in other species (like in Introduction), but a clear link with F. exsecta with hypothesis which could explain the results of the present study, is not present, which makes the Discussion quite deconsolidated from the rest of the manuscript.

8. I am not convinced that speaking about alcoholism in the Introduction is very relevant, as the authors donnot establish a link or make any conclusion related to it in their Discussion.

9. Finally, all along the manuscript, several expressions and sentences are repeated, for instance :

- "research devoted to"

- "The nests used in our experiments (in total 8 mounds) were all situated in dry, sunny places at the southern side of clumps of young pine trees." (the same sentence appears in lines 179 and 221)

10. A detail : is it better to use "spectre" or "spectrum" ?

Reviewer 2 Report

Please see attached PDF file.

The quality of the English in the paper is good. In my review, I have specified some places where a few grammar errors from the paper should be corrected.

Reviewer 3 Report

The manuscript by Korczyńska et.al. investigates the effects of ethanol on ant behavior in field conditions by comparing them with the effects of exposure to water and to acetic acid presented to the ants on small cotton pads soaked in the tested solution. The authors found that exposure to acetic acid evokes an aversive behavioral response whereas ethanol induces antennal exploration of its source and subsequent self-grooming of the antennae. The authors further found that both these olfactory stimuli evoke a mild form of aggressive behavioral response towards their nest mates.

The manuscript is well written. The experiments appear, in general, well-executed and their interpretation is overall fine. I have some suggestions that I would kindly ask the authors to address before I can recommend the manuscript for publication.

1.    Withdrawal from the cotton pad soaked with acetic acid is significant when compared with water or ethanol. The concentration of the acetic acid tested in this experiment evokes an aversive response. Is it possible to test a lower concentration of acetic acid and show whether this behavior is modulated by the concentration of the odorant. The aversiveness of a specific odorant could be due to the concentration used in the experiment. Ants become aversive to the smell of acetic acid and as a result a significant increase in bypassing has been observed. The number of events of bypassing to acetic acid is significantly higher when compared with water or ethanol. Again, testing a lower concentration of acetic acid can shed light on this. Adding one lower concentration of acetic acid will help us to understand this behavioral phenotype better. It would be interesting to know whether acetic acid also evokes an aversive response in lower concentrations. If the experiment is not doable, I request authors to discuss how concentration dependent behavioral modulation could play a role in the discussion.

2.    Was the cotton pad soaked with ethanol, acetic acid or water immediately placed in the experimental arena or was there time between adding the reagents and placing them in the arena? I request authors to mention this in methods section.

3.    The concentration gradient of the odorant used in this study will not remain same throughout the whole time period of the experiment (10:00-17:00). It is therefore interesting to know whether aversiveness and attractiveness to acetic acid and ethanol is more predominant at the early stage of the experiment or it remains same throughout the experiment. For example, is it possible to plot the number of events of bypassing acetic acid at different time intervals.

4.    Various behavioral categories have been reported in this manuscript. I am curious to know whether authors have recorded them as discrete events or if there were some behavioral sequences. Additionally, it would be interesting to know whether these sequences are linked to exposure to specific odorants. When exposed to ethanol, ants show increased self-grooming of the antennae. It would be interesting to know whether that could influence the following social behavior. Extensive self-grooming of the antennae (as being shown in figure 9 and figure 11) could increase sensory acuity and that could alter behavior.

5.    The authors reported an interesting behavioral phenotype where acetic acid becomes aversive and ethanol induces antennal exploration of its source and subsequent self-grooming of the antennae. Both ethanol and acetic acid evoke a mild form of aggressive behavior directed by the tested ant workers to their nest mates. Both these odorants are encoded by different sets of chemosensory receptors and I believe their downstream targets are different. Surprisingly, they evoke a similar behavioral response. I request the  authors provide some explanation.

Round 2

Reviewer 1 Report

Reviewer 1 : Responses to the authors after re-reading

General comments : I thank the authors for having taken into account my comments. After reading the revised manuscript and authors' comments, now I perfectly understood the main goal, the general context and the interest/perspectives of the present study. The subsetting of Introduction in named sections and the added explanations in Discussion helped a lot. It is more : in Results and in other sections, the answers of authors made me get aware that I had made some comments that deal more with my own understanding than a real problem of organisation. Indeed, I proposed the Results were summarized in tables rather than figures, but authors convinced me that given the number of studied behavioural parameters, tables would be very large and unconfortable to read, and would force readers to switch very often between each table to understand the global effects of a tested substance on a defined behaviour. Thus, thinking it well, I have to aknowledge that tables would be quite... indigestible. To resume, the manuscript now removed the doubts I had for the first version, is more undestandable for non-expert readers and bring useful explanations and elements around the interest, the global context and some further perspectives. There come my few comments :

Authors : Actually, we investigated even more parameters used by us to quantify the behaviour observed in workers of F. exsecta exposed to water, ethanol or acetic acid. We analysed 16 behavioural categories, and for each category we calculated three variables: the rate of the occurrence of the tests during which the behaviour in question was observed, the latency from the start of the test to the first observation of that behaviour, and the number of observations of that behaviour made during the test. As a consequence, we analysed the values of 16 x 3 = 48 variables. Additionally, in the case of three behavioural categories we also compared the results obtained for different experimental groups taking into account only these tests during which the analysed behaviour was expressed. In these additional statistical analyses we took into account only two variables per behavioural category, the latency from the start of the test to the first observation of the analyzed behaviour, and the number of observations of that behaviour recorded during the test, as the rate of the occurrence of the tests during which the behaviour in question was observed was always equal to one. Therefore, we made six such additional tests, and in total 54 statistical tests performed to analyse the results of this study.

Yes, that is what I had understood in the first manuscript and I was quite impressed by all this "bunch" of parameters the authors studied.

Authors : This is a misunderstanding. We did not indicate that the tests included in the main experiment reported in the present study were videorecorded. Our methods of recording ant behaviour have been described in the Materials & Methods (lines 236-243) in the following way: „We noted occurrence of various behavioural events classified into 16 behavioural categories quantifying the responses of the ants to the experimental cotton pad and its surroundings, and including their responses to other nestmate ants. We also recorded the latencies from the start of the test to the first observation of each behaviour pattern. Each test was carried out by two persons. The first one acted as an observer and reported the observed behaviour patterns, and the second one noted these informations together with the timing of the first occurrence of each behaviour measured by means of a stopwatch activated at the start of each test.” Videorecording of ant behaviour was mentioned by us solely in the description of other tests that were carried out in our laboratory prior to the present field study. These tests were mentioned in our present paper because their results helped us to choose the concentrations of our experimental compounds (see also the answer to the next point).

Indeed, it is a misunderstanding and I sincerely apologize for this. In addition, I did not think about this : authors are right when they say that field videorecording would be very difficult given the landscape in which they performed their experiments, which would not give good quality videos.